# OpenReview forum: "The Oversight Game: Learning to Cooperatively Balance an AI Agent's Safety and Autonomy"
_ICML.cc/2026/Conference — ICML 2026 regular_

### Official Review · Reviewer_mXMP · 2026-03-08

**Soundness:** 3
**Presentation:** 3
**Significance:** 3
**Originality:** 3
**Overall Recommendation:** 4
**Confidence:** 4

**Summary:**

This paper introduces a framework for AI control that models the interaction between an LLM agent and a human overseer as a two-player game. The authors frame the oversight problems as a coordination problem, where both parties’ objective is to learn learn when oversight is needed. The authors provide two main results, where under the game structure, the LLM agent’s choice to “play” is coupled with human’s utility, and the authors prove the existence of a safe policy for the LLM agent and human that is also “stable” by their definition of deviations. The authors demonstrate the framework on a gridworld setup and on an agentic tool-use tasks setup.

**Compliance With Llm Reviewing Policy:**

Affirmed.

**Final Justification:**

I think the arguments of reviewers uLRF and M6hT about not finding the results generalizable or the assumptions being too strict are valid. I am happy to keep my score as weak accept, but I am fine with rejecting this paper as well based on their reviews and concerns.

**Key Questions For Authors:**

Questions:

1- The shared reward formulation (Equation 6) requires the designer to specify the unsafe action sets for each state. Is it possible to have a “soft version” of this formulation where some actions are unsafe, but with a constraint on a multi-objective setting?

2- The results in Theorem 4.5 show the existence of an equilibrium point, but it may be hard or undecidable to reach this point. I am aware that finding such points is usually intractable on problems with MDPs or Markov games, but the authors could comment on how such an equilibrium point can be achieved, or whether it’s possible to detect such a point in a local region.

3- In the LLM experiments, both the AI and human policies are instantiated as the same 30B-parameter model with different LoRA adapters. Does this shared model architecture and pretraining create an implicit coordination channel that wouldn't exist with a real human overseer? It may be a good idea to train adapters with different base LLMs.

4- Is it possible to extend this setting with continuous action spaces?

**Limitations:**

yes

**Strengths And Weaknesses:**

Strengths:

I like the paper’s setup, where instead of dealing with a fully rational or adversarial agent, they frame the problem as a coordination problem in a two-player setting, and the result of the local alignment theorem is valuable as it deals with a more realistic setting and is more suitable for optimization.

The authors also fine-tune two LoRA adapters on two LLM agents to determine whether their theoretical results hold on a practical setting, even though it deals with two LLMs instead of a human overseer and an LLM agent. I also appreciate the extensive appendix with detailed proofs and how the authors set up the Markov game in detail.

Weaknesses:

The main result of the paper (Theorem 4.2) relies on the assumption defined in Definition 4.1, where for every state in the Markov game, the human’s so-called dummy term does not decrease when the LLM acts autonomously, which is probably too restrictive, even though the authors demonstrate settings where this assumption holds in the appendix.

The framework assumes the designer can correctly specify the set of unsafe actions, the violation penalty, and the interaction costs, which is probably too restrictive for any MDP- or Markov game-based setup in practice. I think it’s possible that the paper’s setting may still be valid where you have a “safe” specification where you can specify the critical safety properties exactly, and have soft or multi-objective constraints on the definition of unsafe actions.

Finally, the main experiments of the paper are done by 2 fine-tuned LLMs, and even though the theoretical results do make sense, this framework may still not hold for actual human operators, as the LLM’s decisions to act autonomously may increase or decrease an unknown utility function that is hard to estimate.

---

> ### Author Rebuttal · Authors · 2026-03-30
>
> We thank Reviewer mXMP for the detailed and constructive review. We address each question below.
>
> **Q1: Soft safety specification / multi-objective formulation:**
>
> Yes, this is a natural and important extension. The shared reward (Eq. 6) can be modified to incorporate probabilistic or graded safety:
>
> $$R_\Phi(s, a_{AI}, a_H) = -\sum_k \lambda_k \cdot p_k(s, a_{exec}) - c_{ask} \cdot 1\{a_{AI} = ask\} - c_{over} \cdot 1\{a_H = oversee\}$$
>
> where $p_k(s, a_{exec})$ represents the probability of violating safety property $k$ and $\lambda_k$ is its severity weight. This remains a team game (and hence an MPG) as long as both players share this reward, so Theorem 4.5 still applies. The key theoretical requirement is that the reward function is common to both players, not that safety is binary.
>
> **Q2: Reachability of the equilibrium:**
>
> Independent Policy Gradient converges to $\varepsilon$-Nash equilibria at polynomial rates in MPGs (Leonardos et al., 2022); we survey additional convergence results in Appendix A.4. Both our gridworld and LLM experiments directly demonstrate this convergence and reachability. More generally, detecting proximity to equilibrium reduces to checking whether any unilateral deviation improves the deviating player's return; if no such improvement exceeds $\varepsilon$, the current joint policy is an $\varepsilon$-Nash equilibrium by definition. The computational cost of this check depends on the state and action space sizes, but the binary action structure of the oversight interface keeps the per-state cost minimal.
>
> **Q3: Shared base model creating an implicit coordination channel:**
>
> This is an important concern. **We conducted an additional experiment using Qwen3-30B-A3B for the AI policy and Llama-3.3-70B-Instruct for the Human policy**, a completely different model family. Coordination still emerges: at risky states, averaging across 5 seeds, the AI ask rate reaches 90.9% and the Human oversee rate reaches 97.3%, comparable to the same-model setting (99.4% and 99.8% respectively). At safe states, both rates remain low (22.3% ask, 8.9% oversee). The best batch reward (+7.48 $\pm$ 4.93) is comparable to the same-model setting (+11.28 $\pm$ 1.71). The safety-critical behavior, deferring at risky states, is preserved across model families, consistent with the inherent theoretical prediction that coordination arises from the shared reward structure rather than from implicit shared representations.
>
> **Q4: Continuous action spaces:**
>
> Potential games themselves accommodate continuous action spaces, but our framework only requires the discrete two-action oversight interface (play/ask and trust/oversee). The base policy $\sigma$ can operate over arbitrary continuous action spaces with no effect on the oversight game wrapper, which gates $\sigma$'s outputs rather than replacing them. We also note that if the reviewer's concern extends to continuous *state* spaces, Ding et al. (2022) establish $O(1/\varepsilon^2)$ iteration complexity that does not explicitly depend on state space size, along with $O(1/\varepsilon^5)$ sample complexity bounds under function approximation, so the convergence guarantees scale to large or continuous state spaces as well.
>
> **Concluding remarks.**
> Given the reviewer's positive assessment of the paper and taking into account the responses and new results above, **we respectfully ask whether these warrant an increase in the overall score.**

---

> > ### Author Rebuttal · Reviewer_mXMP · 2026-04-02
> >
> > I thank the authors did address my comments well, but I will wait on the two reviewers assessment with the reject score before making a change in my score.

---

> > > ### Author Response · Authors · 2026-04-04
> > >
> > > We thank the reviewer for acknowledging that our responses fully resolved their concerns. We also want to thank the reviewer again for suggesting the cross-model experiment (Q3). We believe this is a strong addition to the paper, surfaced thanks to the reviewer during the rebuttal period, and we plan to include it in the camera-ready.
> > >
> > > We wanted to provide a brief update on the status of the other reviews, given the fact you wanted to wait for the assessments of the two reviewers with reject scores. **Reviewer FPs5 moved from 2 to 4 following our rebuttal** and a clarifying follow-up exchange. **Reviewer uLRF moved from 2 to 3.**
> > >
> > > Given that the reviewer's concerns are marked as fully resolved, that the reviewer's original assessment recognized the value of our work, and that the **overall consensus appears to be converging positively, we hope the reviewer now feels comfortable making a change to their score at this point.**

---

### Official Review · Reviewer_FPs5 · 2026-03-12

**Soundness:** 3
**Presentation:** 2
**Significance:** 2
**Originality:** 3
**Overall Recommendation:** 4
**Confidence:** 3

**Summary:**

The model an interaction between a model and a human verifier, where the model is fixed but within an AI wrapper that may decide on each instance whether to ask for help or simply taking the base action. Meanwhile the human may decide whether to trust the model or oversee. The human only overrides the decision of the model asks for help, and the human oversees the model. Under this framework, if the interaction is a Markov potential game and the ask-burden assumption is satisfied, then any local change by an AI model from Ask to Play is locally good for the human. The authors provide an empirical demonstration of this collaboration in a gridworld and an agent tool-use task.

**Compliance With Llm Reviewing Policy:**

Affirmed.

**Final Justification:**

Thanks for the clarification

**Key Questions For Authors:**

What happens experimentally when you increase the cost of overseeing c_over? What motivated the choice of c_over in your experiments? One real-world concern is that the cost of human oversight in practice is really high.

**Limitations:**

Yes

**Strengths And Weaknesses:**

The paper is technically sound and well-written. While I am not fully familiar with all the related literature, I felt the problem to be well-founded and the experimental design to be natural. Overall this is an important area to study.
However, I believe the paper's significance is limited by the strength of the assumptions and scope of the theoretical results. The authors prove results under the assumption that the oversight game is a Markov potential game, which I did not feel was sufficiently justified. I also feel that the discussion of weakening this assumption in section 4.1 is a bit misleading; the authors write that MPG structure and the ask-burden assumption are restrictive assumptions, but that alignment guarantees still hold "these conditions are relaxed." But the result being referenced relaxes the ask-burden assumption while actually strengthening the Markov potential game assumption to a Markov team game assumption.
I also found the discussion of Theorem B.1 to be unsatisfying; I was hoping for a “non-local alignment” theorem showing that if any modification of any number of simultaneous modifications from Ask to Play is beneficial for the AI, then it is also weakly beneficial for the human. Have the authors explored in this direction?

---

> ### Author Rebuttal · Authors · 2026-03-30
>
> We thank Reviewer FPs5 for the thoughtful critique. We address each concern below.
>
> **On the presentation of relaxations in Section 4.1:**
>
> The reviewer raises a fair point about clarity, and we will revise the presentation. To be precise: Proposition 4.4 (perturbed MTG) is a genuine relaxation of *both* conditions, it requires neither exact MPG structure nor the ask-burden assumption, only that rewards are approximately shared (bounded perturbations). The approximate alignment bound $4\kappa/(1-\gamma)$ follows without any additional structural assumptions. Where the reviewer's concern does apply is Section 4.2, which strengthens the game structure to a Markov Team Game (shared reward) in order to obtain the *additional* equilibrium guarantees of Theorem 4.5. We will revise Section 4.1 to present the three settings as distinct operating points with different assumption-guarantee tradeoffs:
>
> | Setting | Assumptions | Guarantee |
> |---------|------------|-----------|
> | MPG + Ask-Burden | General MPG, Def. 4.1 | Exact local alignment (Thm 4.2) |
> | Perturbed MTG | Near-shared reward (weaker) | Approximate alignment (Prop 4.4) |
> | Markov Team Game | Shared reward (stronger) | Exact alignment + equilibrium (Thm 4.5) |
>
> We hope this makes the tradeoff structure transparent and clarifies that Proposition 4.4 is a strict relaxation, while Theorem 4.5 obtains stronger guarantees under stronger assumptions.
>
> **On the MPG assumption being "insufficiently justified":**
>
> We want to clarify that the MPG structure is not an empirical assumption we ask the reader to take on faith, it is a *design specification* that the system builder enforces. The paper's contribution is identifying that if the oversight interface is configured as an MPG, then alignment guarantees follow. This is analogous to mechanism design in economics, Similarly, we do not claim that arbitrary AI-human interactions are MPGs. We specify a reward structure (safety violations, interaction costs) under which the oversight game provably has MPG structure, or a relaxation, and then derive consequences.
>
> **On non-local alignment (simultaneous multi-state deviations):**
>
> We believe a fully general "bundle" analogue of Theorem B.1 is unlikely to follow from our current assumptions alone. Theorem B.1 is pathwise: it guarantees monotonicity along any sequence of single-state ask-to-play deviations that are each individually AI-improving from the start state. This already yields a multi-state extension whenever the simultaneous deviation admits such an ordering. By contrast, a bundled deviation can be AI-beneficial in aggregate because of changes in state visitation, even when some constituent local changes are not individually beneficial, so an unrestricted non-local theorem would likely require stronger structure (e.g., a team game / shared-reward setting, where it is immediate, or a stronger setwise ask-burden condition on the human dummy term).
>
> **On the cost of oversight ($c_{over}$):**
>
> We have conducted a systematic sensitivity analysis sweeping $c_{over}$ across two orders of magnitude while holding $c_{ask} = 0.05$ fixed. The results reveal a robust and gracefully degrading safety profile:
>
> | $c_{over}$ | Violation Rate | AI Ask Rate | H Oversee Rate |
> |-----------|---------------|-------------|----------------|
> | 0.05 | 0.000 | 38.7% | 41.4% |
> | 0.25 | 0.000 | 39.7% | 33.7% |
> | 1.00 | 0.000 | 44.4% | 31.0% |
> | 5.00 | 0.000 | 52.1% | 27.1% |
> | 10.0 | 0.034 | 54.4% | 19.0% |
> | 30.0 | 0.146 | 49.7% | 7.0% |
> | 50.0 | 0.250 | 38.3% | 0.0% |
>
> Safety is preserved (zero violations) across a range of oversight costs ($c_{over} \in [0.05, 5.0]$). As one would expect, the H Oversee rate drops monotonically as $c_{over}$ increases. Violations increase smoothly (3.4% at $c_{over}=10$, 14.6% at 30, 25% at 50) rather than exhibiting a sharp failure.
>
> The AI ask rate exhibits a non-monotonic pattern that reflects adaptive coordination across two regimes. In the rising phase ($c_{over} = 0.05 \to 10$), as the human oversees less frequently, the AI compensates by asking at more states, casting a wider net so that even with sparser human attention, the critical states still get covered. The AI is hedging: if the human only oversees occasionally, the AI should ask broadly near danger to maximize the chance that critical states receive oversight. In the falling phase ($c_{over} = 10 \to 50$), once the human's oversee rate drops sufficiently low, asking becomes increasingly futile. Each ask costs the AI $c_{ask} = 0.05$ with diminishing probability that the human will actually respond.
>
> **Concluding remarks:** We hope the above addresses the reviewer's concerns. Given that the reviewer found the paper technically sound, well-written, and in an important area, **we respectfully ask whether these clarifications and new results warrant a positive adjustment to the overall score.**

---

> > ### Author Rebuttal · Reviewer_FPs5 · 2026-04-03
> >
> > I am confused about your response about Markov team games. If proposition 4.4 is not about team games, then what is a "perturbed MTG"? Is the T not for team? Why is it not denoted as PMPG in the paper and in your rebuttal response? Thanks in advance for the clarification.

---

> > > ### Author Response · Authors · 2026-04-03
> > >
> > > Thanks for the engagement and specific question.
> > >
> > > To clarify: the "T" in "perturbed MTG" does stand for Team, but a perturbed Markov Team Game is *not* a Markov Team Game. The terminology comes from Guo et al. (2025), "Markov $\alpha$-Potential Games," who introduce this class as a strict relaxation. In a Markov Team Game, all players share an identical reward. In a *perturbed* Markov Team Game (PMTG), each player's reward decomposes as $R_i(s,a) = r(s,a) + \xi_i(s,a)$, where $r$ is a shared component and $\xi_i$ is a bounded private perturbation with $|\xi_i| \leq \kappa$. When $\kappa = 0$ this recovers an exact team game; for general $\kappa$ the PMTG is an $\alpha$-MPG with $\alpha \leq 2\kappa/(1-\gamma)^2$.
> > >
> > > The hierarchy of inclusions is:
> > >
> > > Markov Team Games $\subset$ Markov Potential Games $\subset$ Perturbed Markov Team Games $\subset$ Markov $\alpha$-Potential Games
> > >
> > > Our Proposition 4.4 operates at the PMTG level, which is strictly more general than MPGs. Thus we relax *both* the exact MPG assumption and the ask-burden assumption simultaneously, obtaining an approximate alignment bound of $4\kappa/(1-\gamma)$. We agree the terminology is confusing and will clarify this hierarchy explicitly in the camera-ready.
> > >
> > > We refer the reviewer to Section 4.3 of Guo et al. (2025) for the formal definitions and proofs that PMTGs are $\alpha$-MPGs: https://arxiv.org/pdf/2305.12553
> > >
> > > As this is our final opportunity to respond, we hope and believe that we have addressed each of the reviewer's concerns and clarified the contributions of our work.

---

### Official Review · Reviewer_M6hT · 2026-03-13

**Soundness:** 4
**Presentation:** 4
**Significance:** 3
**Originality:** 3
**Overall Recommendation:** 3
**Confidence:** 3

**Summary:**

The paper presents an alignment framework as a two player (here, human and AI) Markov game where each have two actions. Multiple types of constraints such as ask-burden assumption and shared rewards are considered, each supported by a use-case and theoretical guarantees. Key experimental results are presented under the shared reward setting in both classical and LLM tool use setting. Experiments on relaxing to varying rewards are also presented in the Appendix.

**Compliance With Llm Reviewing Policy:**

Affirmed.

**Final Justification:**

The work is well contextualized with both theoretical and experimental contributions for the two player setting presented clearly. Multiple two player settings (MPG, MTG, perturbed MTG) are considered, some more restrictive than others, but supported by corresponding guarantees. While the problem is of significance and approaching it in this principled manner is of interest, the current results seem to be of little practical value and the experimental claims require more rigor. Additionally, the extension to n-player setting from the rebuttal seems to highlight that the work might face significant challenges in extending beyond the two-player setting. Given the last two points which were both discussed during rebuttal, I have slightly reduced my rating.

**Key Questions For Authors:**

1. In line 366, it is interesting that a) the corrective action took the left action and b) the human action after the left was trust and not oversee when at high risk of going into lava, could the authors elaborate on this?
2. Could the authors detail the potential challenges in extending this beyond two players? In addition, it would also help if the authors can contextualize the results in this paper in terms of the extent of their transferability to n-player setting

**Limitations:**

yes

**Strengths And Weaknesses:**

**Strengths**
1. The work is well contextualized with both theoretical and experimental contributions presented clearly
2. The problem is of significance and is approached in a principled manner
3. Multiple settings (MPG, MTG, perturbed MTG) are considered, some more restrictive than others, but all supported by corresponding guarantees
4. Supporting experimental results are reasonably rigorous

**Weaknesses**
1. it would be great if the paper discusses potential challenges in extending beyond two players

---

> ### Author Rebuttal · Authors · 2026-03-30
>
> We thank Reviewer M6hT for the positive and careful assessment. We address both points below and believe we can fully resolve the single identified weakness.
>
> **Q1: On the corrective action at line 366 (left action, then trust):**
>
> Thank you for highlighting this, we agree this is one of the more interesting emergent behaviors in the gridworld. The behavior arises from the interaction between the random safe-action oversight operator and the base policy's Q-values. When the human oversees, the system executes a random safe action, in this case, moving left. At the *next* state (after moving left), the learned policy switches to play/trust because the base policy's Q-values from this new position naturally direct the agent back toward the goal via safe corridors. This illustrates a key feature of the framework: the human's corrections need not be optimal, even random safe actions can redirect the agent to regions where the base policy operates safely. We view this as a strength of the approach: the human need only identify danger (a weak capability), not solve the task (a strong capability), and yet the coordinated policy still achieves zero violations with task completion.
>
> **Q2 (sole identified weakness): Extension to $n$ players:**
>
> We can fully address this. With multiple AI agents ($A$) and multiple human overseers ($H$), the designer must specify how individual play/ask and trust/oversee decisions aggregate into a collective outcome. Natural choices include "any asks" triggering (where a single agent's uncertainty suffices to invoke oversight), majority vote on either side, or "any oversees" rules for the human side. These aggregation rules simply define how the oversight operator Over($s, a_\sigma$) is activated; the game-theoretic structure is unchanged, and the local alignment guarantee extends directly under any such rule. We now state this formally.
>
> *Formal $n$-player extension.* Let the player set be partitioned as $N = A \cup H$, and suppose the resulting interaction is an $n$-player MPG with state-dependent potential $\Phi_s$. Thus for every player $j \in N$,
>
> $$V_s^j(\pi) = \Phi_s(\pi) + U_s^j(\pi_{-j}).$$
>
> Fix an AI agent $i \in A$, a state $s$, and a baseline joint policy $\pi$. Let $\pi_i^{ask@s}$ and $\pi_i^{play@s}$ denote policies that differ only at state $s$, where agent $i$ chooses ask or play, respectively. Suppose
>
> $$V_s^i(\pi_i^{play@s}, \pi_{-i}) \geq V_s^i(\pi_i^{ask@s}, \pi_{-i}).$$
>
> Then by the MPG decomposition,
>
> $$\Phi_s(\pi_i^{play@s}, \pi_{-i}) \geq \Phi_s(\pi_i^{ask@s}, \pi_{-i}).$$
>
> Now assume that for each human $h \in H$, the dummy term is nondecreasing when agent $i$ switches from ask to play:
>
> $$U_s^h(\pi_i^{play@s}, \pi_{-(h,i)}) \geq U_s^h(\pi_i^{ask@s}, \pi_{-(h,i)}).$$
>
> Then for every human $h \in H$,
>
> $$V_s^h(\pi_i^{play@s}, \pi_{-i}) - V_s^h(\pi_i^{ask@s}, \pi_{-i}) = [\Phi_s(\pi_i^{play@s}, \pi_{-i}) - \Phi_s(\pi_i^{ask@s}, \pi_{-i})] + [U_s^h(\pi_i^{play@s}, \pi_{-(h,i)}) - U_s^h(\pi_i^{ask@s}, \pi_{-(h,i)})] \geq 0.$$
>
> Hence an autonomy-increasing deviation by any single AI agent that improves its own value cannot decrease the value of any human agent. Note the proof is essentially unchanged from the two-player case, the key requirement is that each human's dummy term is weakly higher when the given AI switches from ask to play, which both aggregation rules above naturally satisfy.
>
> The MPG framework and associated convergence guarantees are already stated in $n$-player settings (Leonardos et al., 2022; Ding et al., 2022), so our restriction to two players is for conceptual clarity, not a technical limitation. We can add this formal extension to the camera-ready.
>
> **On the overall assessment:** Given that the sole identified weakness ($n$-player extension) can be fully resolved with the formal argument above, and the reviewer found the work well-contextualized, significant, and principled with rigorous experimental support, **we respectfully ask Reviewer M6hT to consider whether these clarifications warrant an increase in the overall score.**

---

> > ### Author Rebuttal · Reviewer_M6hT · 2026-04-04
> >
> > I thank the authors for their rebuttal.
> >
> > On Q1, unless the authors present more evidence of similar cases, I find the justification of human’s trust action, after the left step (with lava being one step away), as being expected or as a feature of the approach weak.
> >
> > On Q2, I believe the above setting presented is not representative of an n-player setting. It seems to essentially boil down to m-independent two player settings, with m as the number of agents and all humans are effectively one human, given the strong assumption
> >
> > "Assume that for each human $h \in H$, the dummy term is nondecreasing when agent $i$ switches from ask to play:
> > $$U_s^h(\pi_i^{play@s}, \pi_{-(h,i)}) \geq U_s^h(\pi_i^{ask@s}, \pi_{-(h,i)}).$$"
> >
> > While I was willing to give the authors the benefit of doubt on strong assumptions in a two player setting (which could potentially represent a human and their personal agent as an example), I am unable to think of a way to justify the above assumption and makes me rethink the value of the two player setting presented in the manuscript.

---

> > > ### Author Response · Authors · 2026-04-04
> > >
> > > We thank the reviewer for the follow-up. We truly believe the concerns stem from miscommunication rather than weaknesses of the work. We hope the below explanations are convincing, and as this is our last chance to respond, we want to convey our confidence in the work.
> > >
> > > **Q1:** Looking at Figure 2, at the location in question where ask/oversee occurs: as we explain in the Oversight Game Formulation paragraph (lines 322-329),
> > >
> > > "We assume the human can identify danger but does not know the optimal correction, modeling a 'capability-gap' setting where the AI is superior at the base task. Thus we set the oversight operator Over as follows: if the AI asks and the human oversees, the system executes a random safe action."
> > >
> > > Since all four movement actions were safe at that state, the Over operator had a 1/4 chance of picking any direction and happened to pick Left. **But once the agent is in the next state, even though it is adjacent to lava, the base policy's Q-values (learned before lava was introduced) already direct the agent rightward at that position.** **Thus there is no reason to learn ask/oversee at that square: it would incur costs without any safety benefit. It is 100% safe for the human to trust here because the base policy will move the agent away from lava, not into it.**
> > >
> > > The reviewer may wonder why the AI and human learned to do ask/oversee at this point in the first place. Note that the base path (dashed line) following the base policy $\sigma$ would lead into lava eventually. The system must learn to "turn right" before reaching the lava, and doing so earlier is more efficient: at the point where ask/oversee occurs, both Right and Down move toward the goal, giving a 2/4 chance of a goal-directed correction under the random-safe-action operator. Waiting until the agent is directly adjacent to the bottom lava would yield only a 1/3 chance of a goal-directed correction. The learned oversight policy is not just safe but strategically efficient in where it allocates human attention.
> > >
> > > **The human's oversight operator is intentionally minimal to model a capability gap**: we could have specified an operator that computes the shortest safe path, but instead deliberately chose a random-safe-action operator to demonstrate that the Oversight Game achieves zero violations even when the human's corrective ability is weak. **This is a core strength of the Lavaland example: the Human and AI cooperatively learn the most economical allocation of costly oversight while maintaining safety, without requiring optimal corrections from the human.**
> > >
> > > **Q2:** We believe there is a misunderstanding here. **The n-player setting does not reduce to m independent two-player games.** All players share a common potential function depending on the *joint* policy of all n players, and all agents' policies jointly determine state transitions. **This is an n-player setting in the same sense that voting in social choice theory is**: the agents share a **common outcome determined by the collective decision**, and the analysis operates on the joint policy space simultaneously. The collective oversight decision could aggregate through majority voting, unanimous-trust rules, or other mechanisms, but this is certainly not m separate games.
> > >
> > > Regarding the per-human ask-burden condition: this is the same assumption from Definition 4.1 applied to each human individually, and does not imply all humans collapse to one, just as a per-voter preference condition in social choice theory does not collapse all voters to one. In the shared-reward setting (Sec 4.2) it holds trivially since all dummy terms are zero. Moreover, for Prop 4.4, the ask-burden assumption is not needed at all, and this result transfers directly to the n-player setting with $\kappa$ bounding the perturbation across all players (see our response to Reviewer FPs5 for details on PMTGs, which, like MPGs, are originally defined for n players). We hope the reviewer is convinced that extending to n players can proceed in many natural directions depending on the aggregation rule, without requiring any unnatural or stronger assumptions.
> > >
> > > **We also want to emphasize that two-player settings, as the reviewer mentions in terms of humans and personal agents, capture the canonical human-agent oversight scenario.** The seminal Off-Switch Game (Hadfield-Menell et al., 2017) that we reference throughout our work considers only two players. **The Oversight Game already delivers several key extensions beyond the Off-Switch Game**: state-based Markov games rather than single-shot settings, explicit modeling of oversight costs, and relaxation of the assumption of perfectly shared reward.
> > >
> > > We hope in light of these clarifications the reviewer is further convinced of the strengths and merits of this work and will consider reflecting this positively in their score. **We truly believe in the strength of our contributions and we hope this discussion has in fact highlighted the paper's strengths.**

---

### Official Review · Reviewer_uLRF · 2026-03-13

**Soundness:** 3
**Presentation:** 2
**Significance:** 1
**Originality:** 2
**Overall Recommendation:** 3
**Confidence:** 4

**Summary:**

The paper attempts to address the problem of passive loss of control by modeling post-deployment AI oversight as an oversight game (a two-player, simultaneous-move markov potential game). The paper purports to provide an “alignment guarantee” in this setting. Specifically, when the game satisfies a restrictive ask-burden assumption, the resulting incentive for the agent to increase its own autonomy cannot reduce the humans expected utility. The authors provide empirical evidence for this in gridworld and LLM agent setting. I am familiar with this area of research.

**Compliance With Llm Reviewing Policy:**

Affirmed.

**Final Justification:**

Following the rebuttal I am still unconvinced of the papers value and impact.

**Key Questions For Authors:**

Please see the comments raised in strengths and weaknesses

Also, I was confused that you introduce the theory in terms of a deployment time intervention on a fixed pretrained model but then the experiments focus on training systems. How should I understand this?

**Limitations:**

yes

**Strengths And Weaknesses:**

The formal setting is assumed to be a cooperative game (motivated by “passive” loss of control risks). I think this is an extremely optimistic assumption. It is reminiscent of approaches to cooperative inverse RL and assistance games [1] (Similar to the off-switch game you discuss.) In terms of formal models of corrigibility, I think this is the most recent and best work [2].

Strengths

The proofs appear to be sound and well written.

I appreciate that the authors clearly have spent time engaging with the relevant literature. But I think the discussion is too broad / high-level and should be more focused (e.g., AI control cf Greenblatt is a substantially different and more specific notion of control than the generic loss of control scenario).

The connection to scalable oversight felt tenuous and could be cut for space.

Figure 1 provides a clear summary of the paper.

Weaknesses

My main concern is that the core results in the paper are not that technically insightful or practically useful.

What is theorem 4.2 really saying? My interpretation is:
Assume that the AI and Human are share some reward (the potential component)
Assume (4.1) that the human does not lose private utility if the AI acts autonomously without “asking”
Assume that the players are rational then the human’s utility cannot decrease by giving the AI more autonomy (except by decreasing the shared potential, but the AI wants to increase this so that doesn't happen).

This theorem feels weak. The assumptions appear to bake in the conclusion. And the assumptions (of shared reward) are overly optimistic. Although the authors do make effort to relax their assumptions (e.g. to perturbed Markov team games).

More simply, my strawman of the result is “If the AI and human are cooperative then they cooperate”


[1] https://people.eecs.berkeley.edu/~russell/papers/neurips20ws-assistance
[2] https://arxiv.org/abs/2305.19861

---

> ### Author Rebuttal · Authors · 2026-03-30
>
> We thank the reviewer for engaging with the paper. We address the concerns below.
>
> **On the Theoretical Framework**
>
> The reviewer argues that the assumptions "bake in the conclusion." We believe there is a misunderstanding here: the paper is a *design framework*, not a descriptive claim. We do not assume AI–human interactions are naturally cooperative. We show that if a designer *engineers* the oversight interface to satisfy specific structural conditions, alignment guarantees follow. The nontrivial content is identifying *which* conditions suffice, and showing they are practically achievable. This is standard practice in mechanism design: one specifies the rules of the game to guarantee desired properties.
>
> The reviewer describes the cooperative game formulation as "extremely optimistic" and draws a comparison to CIRL and assistance games. The cooperative structure is not an assumption about the AI's internal disposition; it is a property *engineered into the oversight wrapper* by the system designer. The base policy σ is treated as an opaque black box that may be arbitrarily misaligned. Unlike CIRL and assistance games, which require the base agent to be designed from the ground up with cooperative objectives, our framework takes any pretrained policy and wraps it in a minimal control layer whose reward structure the designer fully controls. The cooperative structure lives in this thin wrapper, not in the base agent.
>
> **On relaxing assumptions**
>
> We appreciate the reviewer noting our efforts to relax assumptions, and want to emphasize the strength of these results. Proposition 4.4 establishes approximate local alignment in perturbed Markov team games (PMTGs), with an explicit bound of 4κ/(1−γ) on the human's worst-case value loss. This is a precise, quantitative guarantee under strictly weaker conditions that do not require shared rewards or the ask-burden assumption.
>
> Furthermore, Appendix F provides a full experimental demonstration in the Lavaland gridworld under asymmetric costs (c_ask = 0.05, c_over = 0.5), showing that agents converge to a zero-violation equilibrium even when the game is not a team game. These relaxations demonstrate the broad applicability of the framework.
>
> **On the relationship between theory and experiments**
>
> This also seems to be a misunderstanding, though we take responsibility for the unclear exposition. The theory and experiments are fully aligned. In the LLM experiments, Claude serves as the fixed, immutable base policy σ: it pre-computes the execution steps for each ToolEmu scenario, producing the base MDP that the oversight layer wraps. The Qwen models fine-tuned with LoRA are the *deployment-time oversight policies* (play/ask and trust/oversee), which are lightweight wrappers that gate σ's actions without modifying it. This is exactly the setup described in Section 3: a frozen pretrained agent wrapped in a trainable control layer.
>
> The analogy to tools like Claude Code is direct: a pretrained LLM is wrapped in a layer that decides when to proceed autonomously versus request human approval, without modifying the underlying model. We will make this wrapper-vs-base distinction more prominent in the revision.
>
> **On AI control and scalable oversight**
>
> We will sharpen the discussion of AI control (Greenblatt et al., 2024) to make clear we address a complementary threat model (passive, not adversarial loss of control).
>
> Regarding scalable oversight, the reviewer suggests this connection is tenuous. We respectfully disagree. Scalable oversight can be defined as the challenge of humans supervising AI systems whose capabilities may exceed their own. The Oversight Game is a natural instantiation of this problem. The base policy σ may exceed human capabilities at the underlying task, yet the human can still engage meaningfully at safety-critical decision points: they need only evaluate whether an action is safe, not replicate σ's full reasoning. This is exactly the capability gap our Lavaland experiments model, where the human cannot navigate to the goal but can identify and avoid lava.
>
> **On significance**
>
> The reviewer rates significance as 1 but does not identify an alternative approach that provides comparable guarantees. The Oversight Game is, to our knowledge, the first framework that (1) provides sufficient conditions under which autonomy-seeking is provably safe for the human, (2) yields a minimum-oversight equilibrium among safe policies, and (3) demonstrates these properties empirically with LLM agents via independent learning. We believe identifying the structural conditions for safe autonomy is practically important as AI agents are increasingly deployed with human-in-the-loop interfaces.
>
> Taking into account the above responses we respectfully ask the reviewer to reconsider their assessment of the work.

---

> > ### Author Rebuttal · Reviewer_uLRF · 2026-04-03
> >
> > I will update my score slightly, but unfortunately I still don't find the main results very compelling, technically insightful or practically useful, even in light of the authors rebuttal.

---

> > > ### Author Response · Authors · 2026-04-07
> > >
> > > We thank the reviewer for engaging with our rebuttal and for adjusting their score. We appreciate the constructive feedback throughout and will incorporate the suggested improvements to the presentation in the camera-ready.

---

### Decision · Program_Chairs · 2026-04-30

**Decision:**

Accept (regular)

**Comment:**

Reviewers agreed that the paper addresses an important problem, and that the theoretical results are technically sound.

A key concern raised by some reviewers, both before and after the rebuttal, is that the assumptions required to obtain the alignment guarantees are quite strong, which may limit how broadly the theory applies in practice. In particular, these assumptions impose constraints on the oversight and reward structure that may not arise naturally in existing systems.

That said, an important clarification made during the rebuttal is that the proposed theory is not intended to apply to an available deployment, but rather to serve as a design specification that can be engineered into a system. Under this interpretation, the assumptions are not unrealistic, but correspond to conditions that system designers could intentionally enforce in order to obtain alignment guarantees. While the practicality of designing suitable reward or oversight mechanisms warrants further discussion and exploration, the paper provides a principled approach to balancing safety and autonomy by appropriately designing a control layer. Within this scope, the contribution is clear.

One remaining concern relates to the rigor of the experimental evaluation, motivated by a counterintuitive movement of the agent observed in the illustrative Lavaland environment, as well as the lack of baselines in the more realistic ToolEmu scenario. The authors provided an explanation for the counterintuitive behavior that the reviewer found plausible, though the reviewer requested more supporting data. While a more systematic analysis would indeed be preferable, the experiments on ToolEmu, together with the additional experiment with relaxed assumptions on Lavaland in Appendix F, provides evidence in support of the paper’s main claim that the proposed approach performs robustly under weak supervision. In the ToolEmu setting, the paper reports a small optimality gap with high “defer” and “oversight” rates in risky states and low “ask” and “oversight” rates in safe states, which is the behavior predicted by the theory.

Overall, the paper offers a novel perspective on alignment through system‑level design choices and contributes to the growing body of work that emphasizes the role of control layers in shaping agent behavior. For these reasons, I believe the work will be of interest to the community.